# Stereochemical Geometries and Photoluminescence in Pseudo-Halido-Zinc(II) Complexes. Structural Comparison between the Corresponding Cadmium(II) Analogs

**Franz A. Mautner [1],\*, Roland C. Fischer [2], Ana Torvisco [2], Nahed M. H. Salem [3], Amber R. Dugas [4], Shelby F. Aaron [4], Sushant P. Sahu [4], Febee R. Louka [4] and Salah S. Massoud [3,4,\*]**

[1] Institut für Physikalische und Theoretische Chemie, Technische Universität Graz, A-8010 Graz, Austria
[2] Institut für Anorganische Chemie, Technische Universität Graz, Stremayrgasse 9/V, A-8010 Graz, Austria; roland.fischer@tugraz.at (R.C.F.); ana.torviscogomez@tugraz.at (A.T.)
[3] Department of Chemistry, Faculty of Science, Alexandria University, Moharam Bey, Alexandria 21511, Egypt; nahed_s@yahoo.com
[4] Department of Chemistry, University of Louisiana at Lafayette, P.O. Box 43700, Lafayette, LA 70504, USA; amber.dugas2@louisiana.edu (A.R.D.); shelby.aaron1@louisiana.edu (S.F.A.); sushant.sahu@louisiana.edu (S.P.S.); febee.louka@louisiana.edu (F.R.L.)
\* Correspondence: mautner@tugraz.at (F.A.M.); ssmassoud@louisiana.edu (S.S.M.);
Tel.: +43-316-4873-8234 (F.A.M.); +1-337-482-5672 (S.S.M.);
Fax: +43-316-4873-8225 (F.A.M.); +1-337-482-5676 (S.S.M.)

**Abstract:** Six pseudohalide zinc(II) containing a variety of *N*-donor auxiliary amines were structurally characterized. These include two mononuclear trigonal bipyramidal [Zn(NTB)(N$_3$)]ClO$_4$·½H$_2$O (**3**) and [Zn(TPA)(NCS)]ClO$_4$ (**4**), two distorted octahedral [Zn(1,8-damnph)$_2$(dca)$_2$] (**5**) and [Zn(8-amq)$_2$(dca)$_2$] (**6a**) as well as two 1D polymeric chains *catena*-[Zn(isq)$_2$(µ$_{1,5}$-dca)$_2$] (**7**) and *catena*-[Zn(*N,N*-Me$_2$en)$_2$(µ$_{1,5}$-dca)]dca (**8**), where NTB = tris(2-benzimidazolylmethyl)amine, TPA = tris(2-pyridylmethyl)amine, 1,8-damnph = 1,8-diaminonaphthalene, 8-amq = 8-amino-quinoline, isq = iso-quinoline (isq) and *N,N*-Me$_2$en = *N,N*-dimethylethylenediamine. In general, with the exception of **6** and **8**, the complexes exhibited luminescence emission in MeOH associated with red shift of the emission maxima, and the strongest visible fluorescence peak was detected at 421 nm ($\lambda_{ex}$ = 330 nm) in the case of Complex **5**.

**Keywords:** zinc; dicyanamide; azide; isothiocyanate; X-ray; luminescence emission

## 1. Introduction

Group 12 of the periodic table and specifically zinc and cadmium, although they have common properties, exhibit different significant aspects. Both form very stable dipositive ions with an ionic radius of 0.88 and 1.09 Å for Zn$^{2+}$ and Cd$^{2+}$, respectively. This of course results in different stabilities in their metal complexes, where in general, those of Zn(II) are more stable. While Zn$^{2+}$ ions are essential for human life as they are involved in many catalytic enzymatic reactions and metalloenzymes as well as in zinc–protein complexes [1–5], the corresponding Cd(II) compounds are known to be highly toxic for humans and animals, and significantly pollutant to the environment [6]. Interestingly, regardless of the fact that Zn$^{2+}$ and Mg$^{2+}$ have almost identical ionic radii and higher Mg$^{2+}$ concentrations in biological fluids, still Zn$^{2+}$ cannot play the same role of Mg$^{2+}$ in biological systems. Six-coordinate, octahedral [7–16] and tetrahedral, T$_d$ [7,8,17–23] as well as five-coordinate (square pyramidal, SP, and trigonal bipyramid, TBP) [9–11,16,23] are the most common geometries in Zn(II) and Cd(II) compounds. However, seven-coordinate Zn(II) and Cd(II) compounds were also isolated and structurally characterized [13,24,25]. Moreover, three-coordinate species imposed by bulky organic co-ligand(s) were also reported [26,27].

In the presence of auxiliary ligands, the two metal ions $Zn^{2+}$ and $Cd^{2+}$ form extensive metal complexes with pseudo-halide ions {$N_3^-$, $NCS^-$ and dca (dicyanamide ion)} with different degrees of nuclearity and polynuclear species as well as coordination polymers (CPs) with various architecture topologies [11–14,28–32]. The three anions $N_3^-$, $NCS^-$ and dca are known to display a wide range of coordination modes through the two terminal bonding sites (end-to-end) and two sites of the same atom (end-on) as well as mixed bonding [8,9,11–14,25,28–35]. In addition, the corresponding dca ion provides the central amide nitrogen for linking metal ions [14,36,37]. These bonding modes have been previously addressed and summarized [14,32,38–41].

Another interesting feature in studying Zn(II) and Cd(II) complexes, particularly those derived from heterocyclic compounds, is their capabilities to furnish photoluminescence emission [15–17,19–23,25,34,35,42–44], which make them very attracting and fascinating compounds for serving as photochemical devices [43–48]. Two fluorescence tunings of Zn(II) complexes in terms of intensity and/or emission maximum, particularly fluorescence enhancement, chelation enhanced fluorescence (CHEF) or reduction (fluorescence quenching), may occur [48–50]. These were correlated to the ability of the fluorophore coordinated ligand to form a π-contact with the metal cation [51]. In general, the CHE effect results from the stabilization of the excited state in poorly emissive ligands upon coordination with metal ions such as $Zn^{2+}$, which tends to freeze the favorable re-emissive conformation resulting from the lowest excited energy states; from the ligand-centered charge transfer (LCT), mainly π–π* ligand to meatal charge transfer (LMCT) [52,53].

Herein, we investigate the interaction of Zn(II) with different pseudo-halides in the presence of flexible and rigid auxiliary ligands incorporating *N*-donor heterocyclic rings with possible ligand-centered charge transfer (π–π* LMCT); a property that is required for observing luminescent emission. The structures of the ligands used in this study include 2-(ethyl-2-pyridyl)-2-methylquinolyl-methylamine (Meepmqa), tris(2-benzimidazolylmethyl)amine (NTB), tris(2-pyridylmethyl)amine (TPA), 1,8-diaminonaphthalene (1,8-damnph), 8-amino-quinoline (8-amq) and isoquinoline (isq) as well as *N,N*-dimethylethylenediamine (*N,N*-Me₂en), which lacks π–π* transition. The structures of these ligands are represented in Scheme 1. The luminescent emission properties of the complexes were also investigated. The synthesized pseudo-halido-zinc(II) complexes obtained from this study and other related literature will be structurally compared with those determined in the corresponding Cd(II) analogs containing the same pseudo-halides and auxiliary ligands.

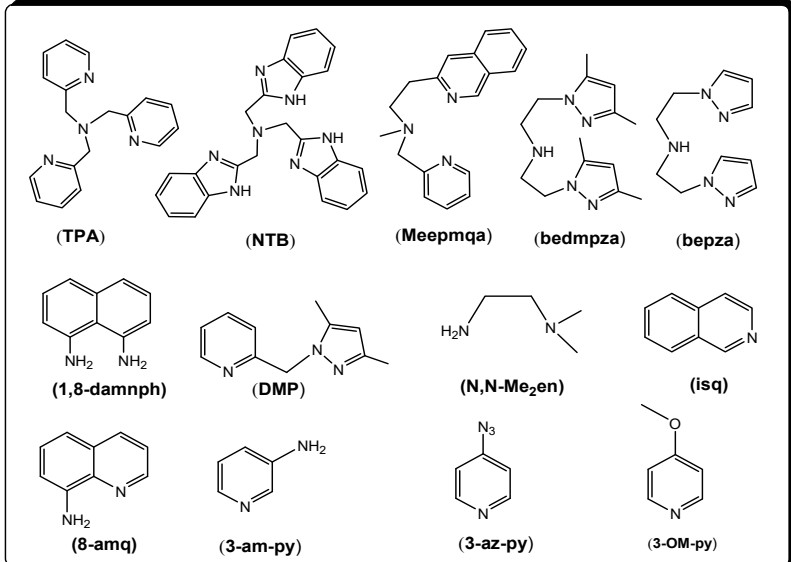

**Scheme 1.** *N*-Donor ligands used in this work and in other related studies.

## 2. Experimental Procedure

### 2.1. Materials and Physical Measurements

8-Aminoquinoline, 1,8-diaminonaphthalene, isoquinoline, 1,2-diaminobenzene, 2,2′,2′′-nitrilotriacetic acid, *N*,*N*-dimethylethylenediamine and 2-picolylamine were purchased from TCI-America (Portland, OR, USA), whereas 2-[2-methylaminoethyl]-pyridine was purchased from Maybridge Co., Belgium. All other chemicals were reagent grade quality. The organic ligands 2-(ethyl-2-pyridyl)-2-methylquinolyl-methylamine (Meepmqa) [28], tris(2-benzimidazolylmethyl)amine (NTB) [54] and tris(2-pyridylmethyl)amine (TPA) [55,56] were prepared according to published procedure. Infrared spectra were recorded on a Cary 630 (ATR-IR, cm$^{-1}$) spectrometer. Electronic spectra were recorded using Agilent 8453 HP diode UV–Vis spectrophotometer (Santa Clara, CA, USA). The fluorescence measurements were performed on Cary Eclipse fluorescence spectrophotometer (Agilent Technologies, Santa Clara, CA, USA). The Fluorescence Quantum Yield (FL QY) of all complexes in methanol was measured by an optically dilute (Optical Density < 0.2) relative method using 1,8 diaminonaphthalene as fluorescent standard of known QY in methanol. The molar conductivities of the complexes were performed in CH$_3$OH solutions at room temperature using Mettler Toledo Seven Easy conductometer (Metler Toledo, Columbus, OH, USA) and calibration by the aid of 1413 µS/cm conductivity standard. The molar conductivity values were determined from $\Lambda_M = (1.0 \times 10^3 \kappa)/[Zn(II)]$, where $\kappa$ is the specific conductance and [Zn(II)] is the molar concentration of the complex. Elemental microanalyses were performed by the Atlantic Microlaboratory, Norcross, GA, USA.

*Caution! Salts of perchlorate and azide as well as their metal complexes are potentially explosive and should be handled with great care and in small quantities.*

### 2.2. Preparation of the Compounds

#### 2.2.1. [Zn(Meepmqa)(N$_3$)$_2$] (**1**)

To a solution containing 0.140 g of 2(ethyl-2-pyridyl)-2-methylquinolyl-methylamine, Meepmqa (0.50 mmol) dissolved in MeOH (20 mL) Zn(ClO$_4$)$_2$·6H$_2$O (0.190 g, 0.50 mmol) was added and this was followed by the addition of NaN$_3$ (0.066 g, 0.5 mmol) dissolved in H$_2$O (1–2 mL). The resulting mixture was heated to boiling for 10 min, filtered while hot and the resulting yellow solution was allowed to stand at room temperature. The yellow crystals, which were separated on the following day, were collected by filtration, washed with propan-2-ol and diethyl ether and air dried (yield: 0.132 g, 62%). Anal. Calcd: C$_{18}$H$_{19}$N$_9$Zn (426.79 g/mol): C, 50.66; H, 4.49; N, 29.54%. Found: C, 50.48; H, 4.33; N, 29.72%. IR bands (ATR, cm$^{-1}$): 3060 (vw), 2916, 2874 (vw) $\nu$(C–H); 2055 (vs) $\nu_a$(N$_3$); 1600 (m), 1571 (m), 1510 (m), 1432 (m) $\nu$(C=C and C=N), 752 (s). UV (CH$_3$OH) $\lambda_{max}$, nm ($\varepsilon_{max}$, M$^{-1}$cm$^{-1}$): 237 (sh), 264 (5.51 × 10$^3$), 272 (5.10 × 10$^3$), 304 (4.51 × 10$^3$), 317 (4.04 × 10$^3$). Molar conductivity, $\Lambda_M$ (MeOH) = 11.3 $\Omega^{-1}$·cm$^2$·mol$^{-1}$.

#### 2.2.2. [Zn(Meepmqa)(dca)]ClO$_4$·½H$_2$O (**2**)

To a mixture containing Zn(ClO$_4$)$_2$·6H$_2$O (0.192 g, 0.5 mmol) and Meepmqa (0.140 g, 0.5 mmol) dissolved in MeOH (15 mL), an aqueous solution (5 mL) of sodium dicyanamide (0.090 g, 1 mmol) was added and the resulting faint yellow solution was heated for 10 min on a steam-bath, filtered through celite and then allowed to crystallize at room temperature. After 2 h, the off-white crystalline compound that separated was collected by filtration, washed with propan-2-ol and Et$_2$O and air dried (yield: 0.23 g, 89%). Characterization: Anal. Calcd: C$_{20}$H$_{20}$ClN$_6$O$_{4.5}$Zn (517.25 g/mol): C, 46.44; H, 3.90; N, 16.25%. Found: C, 46.59, H, 3.82; N, 16.25%. IR bands (ATR, cm$^{-1}$): 3527 $\nu$(O–H); 3130 (vw), 3076 (vw) $\nu$(C–H); 2343 (m), 2268 (m), 2187 (s) (C≡N, dca); 1604 (m), 1514 (w), 1443 (m), 1432 (m), 1380 (m) $\nu$(C=C and C=N); 1076 (vs) $\nu$(Cl–O, ClO$_4^-$). UV (MeOH) $\lambda_{max}$, nm ($\varepsilon_{max}$,

$M^{-1}cm^{-1}$): ~240 (sh), 264 (5.04 × 10³), 272 (4.40 × 10³), 304 (4.32 × 10³), 317 (4.28 × 10³). Molar conductivity, $\Lambda_M$ (MeOH) = 148 $\Omega^{-1}\cdot cm^2\cdot mol^{-1}$.

### 2.2.3. [Zn(NTB)(N₃)]ClO₄·½H₂O (3)

The complex was prepared using a similar procedure as that described for complex **1** except tris(2-benzimidazolylmethyl)amine (NTB) was used instead of TPA (yield: 72%). Characterization: Anal. Calcd: $C_{24}H_{22}ClN_{10}O_{4.5}Zn$ (623.34 g/mol): C, 46.24; H, 3.56; N, 22.47%. Found: C, 46.09; H, 3.61; N, 22.64%. IR bands (ATR, cm⁻¹): 3612 (vw); 3266 (w) ν(N–H); 2062 (vw); 2904 (vw), 2850 (w) 2777 (vw) ν(C–H); 2063 (vs) ν(N₃⁻); 1625 (m), 1596 (m), 1540 (m), 1492 (s), 1472 (m), 1455 (s), 1389 (m), 1343 (m), 1317 (m), 1277 (s) ν(C=C and C=N); 1085 (vs), 1041 (vs) ν(Cl–O, ClO₄⁻); 960 (s), 915 (s), 844 (m) 754 (s), 742 (vs) 619 (vs). UV (MeOH) $\lambda_{max}$, nm ($\varepsilon_{max}$, $M^{-1}cm^{-1}$): 209 (3.68 × 10⁴), 242 (1.22 × 10⁴), 279 (2.04 × 10⁴). $\Lambda_M$ (MeOH) = 143 $\Omega^{-1}\cdot cm^2\cdot mol^{-1}$.

### 2.2.4. [Zn(TPA)(NCS)]ClO₄ (4)

Ammonium thiocyanate, NH₄NCS (0.076 g, 1.0 mmol), was added to a mixture containing tris(2-aminomethylpyridine)amine, TPA (0.146 g, 0.50 mmol) and Zn(ClO₄)₂·6H₂O (0.190 g, 0.5 mmol) dissolved in MeOH (15 mL). The resulting clear solution was heated for 10 min on a steam-bath, filtered and then allowed to crystallize at room temperature. After 3 days, the compound separated was collected by filtration and air dried (yield: 0.215 g, 84%). Colorless single crystals suitable for X-ray analysis were obtained from dilute solution. Characterization: Anal. Calcd: $C_{19}H_{18}ClN_5O_4SZn$ (513.29 g/mol): C, 44.46; H, 3.53; N, 13.64%. Found: C, 43.98, H, 3.51; N, 14.13%. IR bands (ATR, cm⁻¹): 3075 (vw), 2914 (vw) ν(C–H); 2070 (vs) (C≡N, NCS⁻); 1610 (s), 1574 (m), 1483 (m), 1439 (s) ν(C=C and C=N); 1078(vs), 1042 (s), 1026 (s) ν(Cl–O, ClO₄⁻); 756 (vs). $\Lambda_M$ (MeOH) = 148 $\Omega^{-1}\cdot cm^2\cdot mol^{-1}$.

### 2.2.5. [Zn(1,8-damnph)₂(dca)₂] (5)

The complex was prepared using a procedure similar to that described for **2**, except two equivalents of 1,8-damnph were used instead of Meepmqa (yield: 0.203 g, 79%). The same product was obtained with Zn(NO₃)₂·6H₂O. Characterization: Anal. Calcd: $C_{24}H_{20}N_{10}Zn$ (513.87 g/mol): C, 56.09; H, 3.92; N, 27.26%. Found: C, 55.92, H, 3.81; N, 27.01%. IR bands (ATR, cm⁻¹): 3246 (m), 3148 (m) ν(N–H); 3051 (w) ν(C–H); 2260 (s), 2222 (s), 2156 (vs) (C≡N, dca); 1620 (m), 1570 (m), 1397 (m), 1332 (s), 1277 (m) ν(C=C and C=N); 1083 (vs), 998 (s), 911 (m), 814 (s), 757 (vs). UV (MeOH) $\lambda_{max}$, nm ($\varepsilon_{max}$, $M^{-1}cm^{-1}$): 228 (2.82 × 10⁴), ~327 (6.78 × 10³, b). Molar conductivity, $\Lambda_M$ (MeOH) = 9.6 $\Omega^{-1}\cdot cm^2\cdot mol^{-1}$.

### 2.2.6. [Zn(8-amq)₂(dca)]ClO₄ (6)

The complex was prepared using a similar procedure as that described for complex **1** except 8-amq was used instead of Meepmqa (yield: ~67%). Characterization: Anal. Calcd: $C_{20}H_{16}ClN_7Zn$ (519.23 g/mol): C, 46.26; H, 3.11; N, 18.88%. Found: C, 46.08, H, 2.77; N, 20.83%. IR bands (ATR, cm⁻¹): 3248 (m), 3128 (m) ν(N–H); 3075 ν(C–H); 2285 (s), 2235 (s), 2177 (vs) (C≡N, dca); 1628 (m), 1582 (m), 1506 (s), 1474 (m), 1403 (s), 1352 (m), 1323 (s) ν(C=C and C=N); 1070 (vs), 1029 (vs) ν(Cl–O, ClO₄⁻); 827 (s), 787 (s), 764 (m) 717 (s). UV (MeOH) $\lambda_{max}$, nm ($\varepsilon_{max}$, $M^{-1}cm^{-1}$): 241 (2.78 × 10⁴), 265 (sh), 336 (4.74 × 10³). Molar conductivity, $\Lambda_M$ (MeOH) = 147 $\Omega^{-1}\cdot cm^2\cdot mol^{-1}$.

### 2.2.7. *Catena*-[Zn(isq)₂(μ₁,₅-dca)₂] (7)

The complex was prepared using a procedure similar to that described for **2**, except two equivalents of isoquinoline (isq) were used instead of Meepmqa (yield: 77%). Characterization: Anal. Calcd: $C_{22}H_{14}N_8Zn$ (455.79 g/mol): C, 57.97; H, 3.10; N, 24.58%. Found: C, 58.27, H, 3.21; N, 24.59%. IR bands (ATR, cm⁻¹): 3075 (w) ν(C–H); 2290 (s), 2234 (s), 2169 (vs) (C≡N, dca); 1633 (m), 1596 (m), 1501 (m), 1384 (s), 1346 (m), 1275 (s) ν(C=C and C=N);

1042 (s), 1018 (m) 952 (m), 930 (s), 863 (m), 819 (vs), 773 (s) 741 (vs). UV (MeOH, saturated solution) $\lambda_{max}$, nm: 210 (sh), 260, 268, 280, 306, 319.

### 2.2.8. *Catena*-[Zn(N,N-Me₂en)₂(μ₁,₅-dca)]dca (**8**)

The complex was prepared using a procedure similar to that described for **2**, except two equivalents of *N,N*-dimethylethylenediamine (*N,N*-Me₂en) were used instead of Meepmqa (yield: 68%). Characterization: Anal. Calcd: $C_{12}H_{24}N_{10}Zn$ (455.79 g/mol): C, 38.56; H, 6.47; N, 37.47%. Found: C, 38.45, H, 6.28; N, 37.59%. IR bands (ATR, cm⁻¹): 3295 (w), 3249 (w) ν(N–H); 3167 (w), 2967 (vw), 2897 (vw) ν(C–H); 2318 (m), 2284 (m), 2261 (m), 2239 (m), 2168 (vs), 2189 (vs) (C≡N, dca); 1595 (m), 1466 (m), 1387 (m), 1140 (m), 1058 (m), 1006 (m), 930 (m), 878 (m), 283 (m), 657 (m). UV (MeOH, saturated solution) $\lambda_{max}$, nm: ~243 (1.00 × 10⁴), 265 (sh), 271 (1.19 × 10⁴), 278 (9.67 × 10³). Molar conductivity, $\Lambda_M$ (MeOH) = 141 $\Omega^{-1} \cdot cm^2 \cdot mol^{-1}$.

### 2.3. X-Ray Crystal Structure Analysis

The X-ray single-crystal data of the six title compounds were collected on a Bruker-AXS APEX II CCD diffractometer (Bruker AXS, Karlsruhe, Germany) at 100(2) K. The crystallographic data, conditions retained for the intensity data collection and some features of the structure refinements are listed in Table 1. Data collections were performed with Mo-K$\alpha$ radiation ($\lambda$ = 0.71073 Å); data processing, Lorentz-polarization and absorption corrections were performed using the APEX and SADABS computer programs [57,58]. The structures were solved by direct methods and refined by full-matrix least-squares methods on F², using the SHELX program library [59–61]. All non-hydrogen atoms were refined anisotropically. The hydrogen atoms were located from difference Fourier maps, assigned with isotropic displacement factors. Geometrical constraints (HFIX) were applied only for H atoms bonded to C atoms. Further programs used: Mercury and PLATON [62,63].

**Table 1.** Crystallographic data and processing parameters of **3**, **4**, **5**, **6a**, **7** and **8**.

| Compound | 3 | 4 | 5 |
|---|---|---|---|
| Empirical formula | $C_{48}H_{44}Cl_2N_{20}O_9Zn_2$ | $C_{19}H_{18}ClN_5O_4SZn$ | $C_{24}H_{20}N_{10}Zn$ |
| Formula mass | 1246.71 | 513.28 | 513.87 |
| System | Triclinic | Monoclinic | Monoclinic |
| Space group | P-1 | P2₁/c | P2₁/n |
| a (Å) | 13.7520(9) | 15.2869(9) | 7.4176(4) |
| b (Å) | 13.8644(8) | 9.6319(5) | 11.2397(6) |
| c (Å) | 17.3919(10) | 14.7015(9) | 13.3287(6) |
| α (°) | 99.093(3) | 90 | 90 |
| β (°) | 102.308(3) | 91.591(4) | 98.738(3) |
| γ (°) | 119.270(2) | 90 | 90 |
| V (Å³) | 2689.0(3) | 2163.8(2) | 1098.34(10) |
| Z | 2 | 4 | 2 |
| $D_{calc}$ (Mg/m³) | 1.540 | 1.576 | 1.554 |
| θ max (°) | 27.544 | 28.782 | 33.204 |
| Data collected | 24602 | 36586 | 49264 |
| Unique refl./$R_{int}$ | 12326/0.0642 | 5570/0.0853 | 4201/0.1122 |
| Parameters/Restraints | 725/140 | 280/0 | 176/0 |
| Goodness-of-Fit on F² | 1.081 | 1.051 | 1.023 |
| R1/wR2 (all data) | 0.0897/0.2605 | 0.0520/0.1274 | 0.0392/0.0826 |
| Residual extrema (e/Å³) | 3.202/−2.305 | 1.97/−0.73 | 0.59/−0.55 |
| Compound | 6a | 7 | 8 |

| Empirical formula | $C_{22}H_{16}N_{10}Zn$ | $C_{22}H_{14}N_8Zn$ | $C_{36}H_{72}N_{30}Zn_3$ |
|---|---|---|---|
| Formula mass | 485.84 | 455.78 | 1121.34 |
| System | Monoclinic | Triclinic | Monoclinic |
| Space group | $P2_1/n$ | P-1 | C m |
| a (Å) | 8.8263(3) | 7.406(3) | 13.4600(13) |
| b (Å) | 7.2601(3) | 10.549(4) | 26.939(3) |
| c (Å) | 16.0965(6) | 13.314(5) | 7.4731(7) |
| α (°) | 90 | 102.690(14) | 90 |
| β (°) | 94.614(2) | 90.274(17) | 109.599(6) |
| γ (°) | 90 | 110.278(16) | 90 |
| V (Å³) | 1028.12(7) | 948.1(6) | 2522.8(5) |
| Z | 2 | 2 | 2 |
| $D_{calc}$ (Mg/m³) | 1.569 | 1.596 | 1.459 |
| θ max (°) | 33.160 | 33.282 | 30.104 |
| Data collected | 101990 | 76319 | 38697 |
| Unique refl./$R_{int}$ | 3932/0.0740 | 7224/0.0762 | 7333/0.0436 |
| Parameters/Restraints | 159/0 | 283/0 | 364/50 |
| Goodness-of-Fit on $F^2$ | 1.074 | 1.104 | 1.038 |
| R1/wR2 (all data) | 0.0282/0.0745 | 0.0379/0.0899 | 0.0253/0.0637 |
| Residual extrema (e/Å³) | 0.49/−0.44 | 0.60/−0.70 | 0.72/−0.35 |

## 3. Results and Discussion

### 3.1. Synthetic Aspects

The interaction of a methanolic solution of $Zn(NO_3)_2 \cdot 6H_2O$ or $Zn(ClO_4)_2 \cdot 6H_2O$ and mono- or bidentate *N*-donor ligands (L) with an aqueous solution containing the pseudo-halides $NaN_3$, $NH_4NCS$ or Nadca in the stoichiometric ratio 1:1:2 afforded only the corresponding di-pseudohalido compounds, $[Zn(L)_2(X)_2]$ {L = 1,8-damnph, X = dca (**5**); L = isq, X = dca (**7**)}. A polymeric complex with a similar chemical formula, *catena*-[Zn(*N,N*-Me₂en)₂(µ₁,₅-dca]dca (**8**), was obtained from the reaction of $Zn(NO_3)_2 \cdot 6H_2O$, *N,N*-Me₂en and Nadca. On the other hand, under similar conditions the corresponding reaction of $Zn(ClO_4)_2 \cdot 6H_2O$ and 8-amq ligand in the presence of Nadca resulted in the formation of a mixture of the major product $[Zn(8-amq)_2(dca)]ClO_4$ (**6**) and a small amount of the structurally characterized monomeric compound $[Zn(8-amq)_2(dca)_2]$ (**6a**). The elemental microanalysis was close to the proposed formula **6** (Anal. Calcd: $C_{20}H_{16}ClN_7Zn$: C, 46.26; H, 3.11; N, 18.88%. Found: C, 46.08, H, 2.77; N, 20.83%), and the presence of the dca and perchlorate in **6** was confirmed by IR. The molar conductivity measurement of the compound agreed with the 1:1 electrolytic nature of the complex (see Section 2—Experimental Procedure). However, in the case of the tridentate amine, Meepmqa and $Zn(ClO_4)_2$ five-coordinate $[Zn(Meepmqa)(N_3)_2]$ (**1**) and $[Zn(Meepmqa)(dca)]ClO_4 \cdot \frac{1}{2}H_2O$ (**2**) complexes were isolated in the presence of aqueous $NaN_3$ and Nadca solutions, respectively. When $Zn(ClO_4)_2 \cdot 6H_2O$ was employed with the tripod tetraamines (NTB, TPA) and the pseudohalide salts in the 1:1:2 molar ratio, the five-coordinate mono-pseudohalido perchlorate compounds $[Zn(L)(X)]ClO_4$ {L = NTB, X = $N_3^-$ (**3**); L = TPA, X = $NCS^-$ (**4**)} were the only products regardless of the excess pseudohalide salts used. Similar products, $[Zn(NTB)(N_3)]NO_3$ (**3a**, Supplementary Materials (SM)) and $[Zn(TPA)(NCS)]NO_3 \cdot \frac{1}{2}H_2O$ (**4b**, SM), were obtained in the presence of the appropriate co-ligand and pseudohalide salt when zinc(II) nitrate was used instead of the perchlorate (see SM). Whenever the solubility of the complexes in $CH_3OH$ permits, the molar conductivity values, $\Lambda_M$ for **2–4**, **6** and **8**, were found to be in the range 141–148 $\Omega^{-1} \cdot cm^2 \cdot mol^{-1}$, which are in complete agreement for their 1:1 electrolytic nature [64]. The non-electrolytic nature of **1** and **5** was also

confirmed ($\Lambda_M$ = 11.3 and 9.6 $\Omega^{-1}\cdot cm^2\cdot mol^{-1}$, respectively). The compounds were characterized by IR and UV spectroscopy as well as elemental microanalyses. The molecular structures of **3–5** and **6a–8** were determined by single crystal X-ray crystallography.

### 3.2. IR Spectra of the Complexes

The infra-red spectra of the complexes under investigation display general characteristic features for each category of the pseudohalides. The azido compounds **1** and **3** exhibit their strong vibrational frequencies at 2055 and 2063 cm$^{-1}$, respectively, due to the asymmetric stretching vibrations, $\nu_a$, of the azido ligands [8,9,11,28,29,31,34,35,38–40]. The isothiocyanato complex **4** reveals a strong band at 2070 cm$^{-1}$, which is inconsistent with *N*–NCS bonding; the *S*–NCS bonding, in general, absorbs at a frequency value slightly greater than 2100 cm$^{-1}$ [12,33,40,65]. The IR spectra of the dicyanamido complexes **5**, **6** and **7** display three medium–strong intense bands over the 2290–2260, 2240–2220 and 2160–2180 cm$^{-1}$ regions, whereas the corresponding bands were located at higher values for [Zn(Meepa)(dca)]ClO$_4$·½H$_2$O (**2**). These vibrational bands are attributed to $\nu_{as}(C\equiv N)$ and $\nu_s + \nu_{as}(C\equiv N)$ and $\nu_s(C\equiv N)$, respectively [14,25,30,32,36,41,66]. The observed dca bands in *catena*-[Zn(*N,N*-Me$_2$en)$_2$($\mu_{1,5}$-dca)]dca (**8**) were further split into 2318, 2284, 2261, 2239, 2168, and 2189 cm$^{-1}$ as expected because of the presence of dca anions in two different bonding modes. The perchlorate compound **2** reveals its strong vibrational frequency value due to the Cl–O stretching of the counter ClO$_4^-$ ion at 1073 cm$^{-1}$. The split of this band into two in [Zn(8-amq)$_2$(dca)]ClO$_4$ (**6**) or three in [Zn(TPA)(NCS)]ClO$_4$ (**4**) (see Experimental Procedure section) may result from reducing the local symmetry of the ClO$_4^-$ ion from $T_d$ to C$_{3v}$ or C$_{2v}$.

### 3.3. Description of the Structures

#### 3.3.1. [Zn(NTB)(N$_3$)]ClO$_4$·½H$_2$O (3) and [Zn(TPA)(NCS)]ClO$_4$ (4)

The asymmetric unit of compound **3** contains two monomeric [Zn(NTB)(N$_3$)]$^+$ complex cations, one ordered and one disordered perchlorate counter anion and a non-coordinated water molecule, whereas compound **4** crystallizes with one monomeric [Zn(TPA)(NCS)]$^+$ complex cation and one perchlorate counter anion in the asymmetric unit. In **3**, each Zn$^{2+}$ center is penta-coordinated by four *N*-donor atoms of the tripod NTB molecule and *N*-donor of the terminal azide anion, and in **4** by four *N*-donor atoms of the tripod TPA molecule and *N*-coordinated NCS$^-$ anion (Figure 1 and selected bond parameters are shown in Table S1). The ZnN$_5$ polyhedra show distorted trigonal bipyramidal (TBP) geometry with a $\tau$ value of 0.87 [0.83] for Zn1 [Zn2] polyhedron in **3**, and 0.85 in the case of **4** ($\tau$ value refers to 0 for ideal square pyramid SP and 1 for ideal TBP geometry) [67]. The equatorial positions occupied by the coordinated *N*-donor of the three arms of tripod ligands with Zn–N(eq) bond distances are in the range from 2.002(5) to 2.079(3) Å, and N-(eq)-Zn-N(eq) angles from 106.1(2) to 124.32(11)°. One axial position is occupied by the coordinated N-donor atom of the pseudohalide anion with a Zn–N(azide) bond distance of 2.020(5) [2.019(5)] Å in **3** and a Zn–N(NCS) bond distance of 2.006(3) Å in **4**. The other axial position is occupied by the central amine *N*-donor of the tripod NTB (in **3**) or TPA (in **4**) ligand molecule, respectively, with a longer Zn–N(amine) bond length of 2.491(5) [2.489(5)] Å in **3** and 2.261(3) Å in **4**. The axial N–Zn–N bond angle is 174.4(2)° [173.0(2)°] in **3** and 175.26(12)° in **4**. The terminal azide groups are asymmetric with a mean difference of the N–N bond lengths of 0.034 Å. The Zn–N–N and N–N–N bond angles in **3** are 119.9(4)° [120.7(4)°] and 178.5(6)° [177.6(7)°], whereas in **4** the Zn–N–C and N–C–S bond angles are 154.3(3)° and 178.8(3)°. In **3**, the building blocks are interlinked by hydrogen bonds of type O–H···O, N–H···O and N–H···N, whereas in **4** no classical hydrogen bonds are observed (Figures S1 and S2, Table S2).

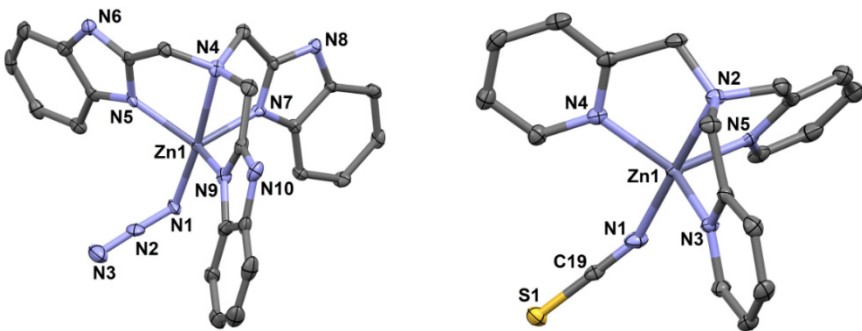

**Figure 1.** Complex cations of Zn1 centers in structures of **3** (**left**) and of **4** (**right**).

### 3.3.2. [Zn(1,8-damnph)2(dca)2] (**5**) and [Zn(8-amq)2(dca)2] (**6a**)

In the centrosymmetric mononuclear complexes [Zn(1,8-damnph)2(dca)2] (**5**) and [Zn(8-amq)2(dca)2] (**6a**) (Figure 2 and selected bond parameters are shown in Table S1), the Zn(II) centers form axially elongated octahedra by the ligation of four *N*-donor atoms of two *trans*-coordinated 1,8-diaminonaphthalene (**5**) and 8-aminoquinoline (**6a**) molecules and of two terminal dicyanamide ligands. The axial Zn1–N(dca) bond distances are 2.2179(14) and 2.2051(11) Å, whereas the equatorial Zn1–N bond distances are in the range from 2.1199(15) to 2.1620(14) Å, respectively. The N1–Zn1–N2 bite angle is 80.25(6) and 78.94(4)° for the six-membered chelate ring in **5** and the five-membered chelate ring in **6a**, respectively, whereas the other N–Zn–N cisoid bond angles deviate less than 2.8° in **5** and 5.9° in **6a** from a rectangular angle. The naphthalene-rings in **5**, as well as the quinoline rings in **6a**, are inclined by 52.2 and 22.3° to their mean equatorial ZnN4 plane. The terminal dca anions have the following bond parameters: Zn–N–C: 161.72(14) and 147.31(10)°; N–C–N: from 171.69(14) to 173.71(14)°; C–N–C: 121.66(16) and 122.26(12)°; C–N(nitrile): from 1.155(2) to 1.1648(18) Å; N–C(amine): from 1.3042(17) to 1.321(2) Å. Hydrogen bonds of type N–H·N to adjacent non-coordinated N5 acceptor atoms of dca anions form a supramolecular 2D system in **6a**, whereas N4 and N5 atoms of neighboring dca anions act as acceptors for hydrogen bonds of type N–H···N to generate a supramolecular 3D network structure with **bcu** topology in **5** (Table S2, Figures S3 and S4) [61–63].

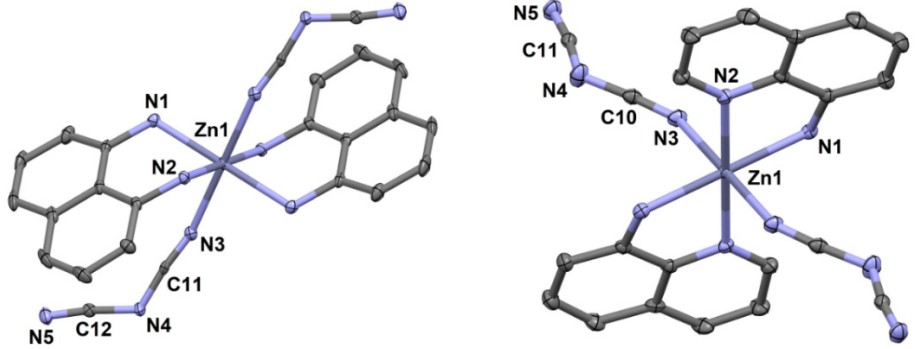

**Figure 2.** Coordination figures of **5** (**left**) and of **6a** (**right**).

### 3.3.3. *Catena*-[Zn(isq)2(μ1,5-dca)2] (**7**) and *Catena*-[Zn(N,N-Me2en)(μ1,5-dca)]dca (**8**)

The common feature of **7** and **8** is 1D polymeric chains (Figures 3 and 4 and bond parameters are shown in Table S1). In **7**, neutral chains of catena-[Zn(isq)2(μ1,5-dca)2] are formed via bis μ1,5-*dca* bridging ligands, which connect the Zn(II) metal centers along the *a*-axis of the unit cell. Each ZnN6 octahedron is completed by two terminal isoquinoline molecules in *trans* disposition. The Zn–N bond distances vary from 2.1369(17) to 2.1735(17) Å and N–Zn–N bond angles deviate less than 0.60° from ideal octahedral

geometry. The parallel stacking of the isoquinoline molecules is stabilized by the formation of π···π ring···ring interactions (Table S3, Figure S5). The cationic chains in **8** are formed via single μ$_{1,5}$-*dca* bridging ligands, which connect the metal centers in an alternate [Zn1···Zn1···Zn2] sequence along the *b*-axis of the unit cell. Each distorted ZnN$_6$ octahedron is completed by four N-donors of two chelating *N,N*-Me$_2$en molecules in *trans* disposition. The Zn–N bond distances vary from 2.063(4) to 2.278(3) Å and N–Zn–N bond angles deviate up to 7.5° from ideal octahedral geometry. Hydrogen bonds of type N–H···N are formed between *N,N*-Me$_2$en and dca counter anions, which are arranged parallel to the μ$_{1,5}$-dca bridging ligands, to generate a supramolecular 2D system oriented along the *a*- and *b*-axis of the unit cell (Table S2, Figure S6). The dca anions in **7** and **8** have the following bond parameters: Zn–N–C: from 148.80(15) to 168.49(16)°; N–C–N: from 170.5(4) to 175.1(2)°; C–N–C: from 119.09(17) to 125.7(4)°; C–N(nitrile): from 1.144(4) to 117.0(5) Å; N–C(amine): from 1.296(4) to 1.313(2) Å. Partial two-fold disorder is observed for *N,N*-Me$_2$en molecules ligated to Zn2, which is located on the special position.

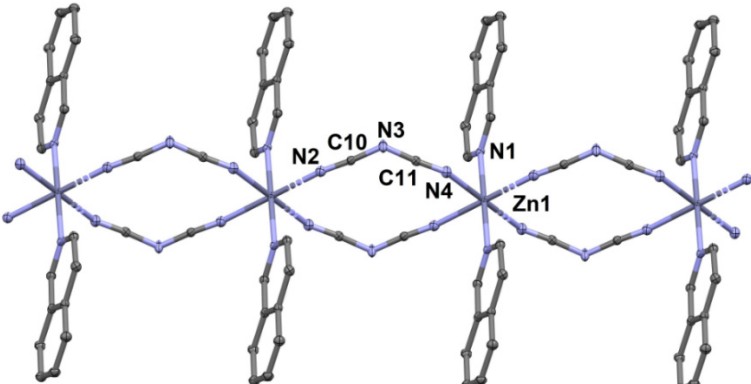

**Figure 3.** Polymeric chain of **7**.

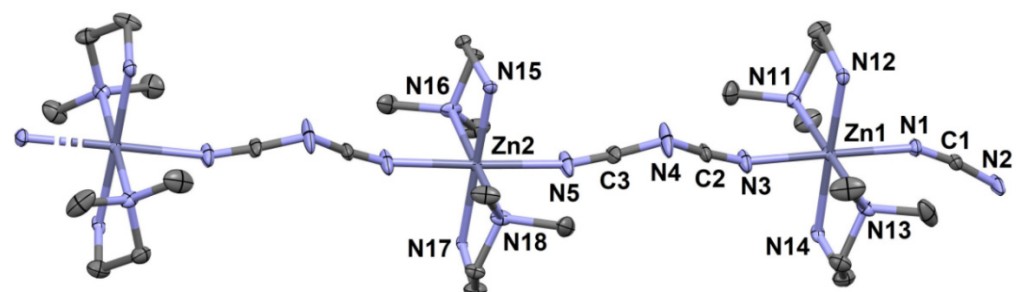

**Figure 4.** Polymeric cationic chain of **8**. Only one orientation of disordered *N,N*-Me$_2$en groups around Zn2 is given for clarity.

### *3.4. Structural Comparison between Zinc(II) Complexes and Cadmium(II) Analogs*

In this section, we are focusing on the Zn(II)-pseudohalide complexes and their corresponding structurally characterized Cd(II) analogs containing the same auxiliary ligands and pseudohalides. These compounds are collected in Table 2 for comparison. Inspection of the data in this table reveals that although the two metal ions can produce compounds with similar geometrical features and the same pseudohalide bonding modes (see entries 7–10, 12 and 13 and 20–25), still many other compounds show significant differences with respect to geometry, nuclearity and pseudohalide bonding modes (entries 1–5, 9 and 11 and 14–19). This behavior precludes any possibility of predicting the structure of any of the two complexes if one of them was determined. This could not only be

attributed to the smaller ionic size of the $Zn^{2+}$ ion but also to its hard Lewis acid nature in regard to the soft $Cd^{2+}$ ion.

**Table 2.** Structural parameters of pseudohalido-Zn(II) complexes and their Cd(II) analogs *.

| # | Zn(II) or Cd(II) Complex | Pseudoh. Bonding | Geom. | Dim./Nuc. | Ref. |
|---|---|---|---|---|---|
| 1 | [Zn(TPA)(N₃)]ClO₄ (**4a**) | monodentate $N_3^-$ | TBP | mononuclear | [68] |
| 2 | [Cd₆(TPA)₄(μ₁,₁,₃-N₃)₄(μ₁,₁N₃)₆](ClO₄)₂·2H₂O | μ₁,₁,₃- $N_3^-$, μ₁,₁- $N_3^-$ | dist. Oₕ, and PBP | hexanuclear | [13] |
| 3 | [Zn(TPA)(N₃)₂] | monodentate $N_3^-$ | dist. Oₕ, | mononuclear | [68] |
| 4 | *catena*-[Zn(N,N-Me₂en)₂(μ₁,₅-dca)]dca (**8**) | μ₁,₅-dca | dist. Oₕ | CPs,1D | This work |
| 5 | *catena*-[Cd(N,N-Me₂en)(μ₁,₅-dca)₂] | μ₁,₅-dca | dist. Oₕ | CPs, 1D double chains | [14] |
| 6 | [Zn(isq)₂(NCS)₂] | monodentate $NCS^-$ | dist. T_d | mononuclear | [69] |
| 7 | *trans*-[Zn(isq)₄(NCS)₂] | monodentate $NCS^-$ | *trans*-Oₕ | mononuclear | [69] |
| 8 | *trans*-[Cd(isq)₄(NCS)₂] | monodentate $NCS^-$ | *trans*-Oₕ | mononuclear | [69] |
| 9 | *cis*-[Zn(8-amq)₂(NCS)₂] | monodentate $NCS^-$ | *cis*-Oₕ | mononuclear | [70] |
| 10 | *cis*-[Cd(8-amq)₂(NCS)₂] | monodentate $NCS^-$ | *cis*-Oₕ | mononuclear | [34] |
| 11 | *catena*-[Cd(8-amq)(μ₁,₃-NCS)₂] | μ₁,₃-$NCS^-$ | dis. Oh | CPs, 1D | [71] |
| 12 | *trans*-[Zn(1,8-damnph)₂(dca)₂] (**5**) | monodentate dca | *trans*-Oh | mononuclear | This work |
| 13 | *trans*-[Cd(1,8-damnph)₂(dca)₂] | monodentate dca | *trans*-Oh | mononuclear | [14] |
| 14 | [Zn(bedmpza)(N₃)]ClO₄ | monodentate $N_3^-$ | dist. TBP | mononuclear | [11] |
| 15 | [Cd(bedmpza)(μ₁,₁-N₃)(N₃)]₂·1.5H₂O | μ₁,₁-$N_3^-$, monodentate $N_3^-$ | dist. Oh | dinuclear | [11] |
| 16 | [Zn(bepza)(NCS)₂] | monodentate $NCS^-$ | dist. TBP | monomer | [12] |
| 17 | [Cd₂(bepza)₂(μ₁,₃-NCS)₂(NCS)₂] | μ₁,₃-$NCS^-$, monodentate $NCS^-$ | dist. Oh | dinuclear | [12] |
| 18 | [Zn(DMP)(μ₁,₁-N₃)(N₃)]₂ | μ₁,₁- $N_3^-$ | dist. TBP | dinuclear | [28] |
| 19 | *catena*-[Cd(DMP)(μ₁,₁-N₃)₂] | μ₁,₁- $N_3^-$ | dist. Oh | CPs, 1D | [28] |
| 20 | *catena*-[Zn(3-ampy)₂(μ₁,₅-dca)₂] | μ₁,₅-dca | dist. Oh | CPs, 3D | [30] |
| 21 | *catena*-[Cd(3-ampy)(μ₁,₃-dca)(μ₁,₅-dca)] | μ₁,₅-dca, μ₁,₃-dca | dist. Oh | CPs, 3D | [30] |
| 22 | *catena*-[Zn(4-OMP)₂(μ₁,₅-dca)₂] | μ₁,₅-dca | dist. Oh | CPs, 1D | [32] |
| 23 | *catena*-[Cd(4-OMP)₂(μ₁,₅-dca)₂] | μ₁,₅-dca | dist. Oh | CPs, 1D | [32] |
| 24 | *catena*-[Zn(4-azpy)₂(μ₁,₁-N₃)(μ₁,₃-N₃)] | μ₁,₁- $N_3^-$, μ₁,₃- $N_3^-$ | dist. Oh | CPs, 1D | [31] |
| 25 | *catena*-[Cd₂(4-azpy)₄(μ₁,₁-N₃)₂(μ₁,₃-N₃)₂] | μ₁,₁- $N_3^-$, μ₁,₃-$N_3^-$ | dist. Oh | CPs, 1D | [31] |

* Abbreviations: Pseudoh. = pseudohalide anion, CPs = coordination polymer, Dim = dimensional, Nuc = nuclearity, dist. = distorted, Oₕ = octahedral, T_d = tetrahedral, SP = square pyramidal, PBP = pentagonal bipyramid, TPA = Tris(2-pyridyl-methy)amine, N,N-Me₂en = N,N-dimethylethylenediamine, 8-amq = 8-aminoquinoline, 1,8-damnph = 1,8-diaminonaph-thalene, isq = isoquinoline, 3-am-py = 3-aminopyridine, 4-az-py = 4-azidopyridine, 4-OM-py = 4-methoxypyridine, DMP = 2-[3,5-dimethyl-*1H*-pyrazol-1ylmethyl)]pyridine, bepza = [bis(2-pyrazol-*1H*-yl)ethyl)]amine, bedmpza = = [bis((3,5-di-methyl)-2-pyrazol-*1H*-yl)ethyl)]amine.

### 3.5. Luminescence Emission

The optical properties of the synthesized pseudo-halido-zinc(II) complexes were investigated in methanol by UV–Vis and luminescence spectroscopy. To examine the effect of the pseudohalide on the luminescence emission, the TPA series of [Zn(TPA)(X)]ClO₄ (X = NCS⁻, **4**; X = N₃⁻, **4a** [68]; X = dca, **4b** [72]) were studied. These complexes, together with containing a TPA ligand, show two absorption bands centered around 205 and 260 nm,

which can be assigned to ligand-centered π–π* and n–π* transitions of the pyridyl rings and those containing N atoms. As seen in Figure 5, upon photoexcitation at 260 nm, complexes **4**, **4a** and **4b** showed pronounced emission maxima at ~450 nm and a large red shift up to Δλ = 190 nm was observed, which can be ascribed to ligand to sensitized charge transfer transition (LCT) from the lowest excited energy states within the whole complex. Interestingly, parent ligand-centered charge transfer (LCT) emission in the azido complex **4a** was quenched in **4** and **4b**. In addition, it is noted from normalized emission intensities that complex **4a** with $N_3^-$ exhibited the strongest luminescence at ~450 nm among the TPA series (Figure 5, left). The luminescence of **4a** is found to be nearly 1.6 times greater than the visible luminescence of complex **4b** with dca.

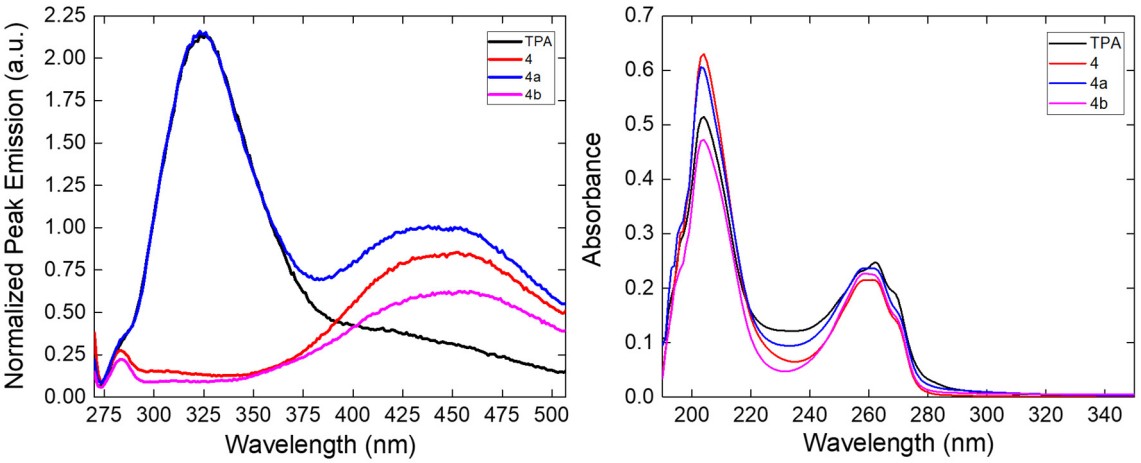

**Figure 5.** (**Left**): the normalized emission intensities (represented in reference to complex **4a**, which exhibited highest peak intensity) at around 450 nm at $λ_{ex}$ = 260 nm, and (**Right**): UV–Vis. absorption spectra for pseudohalido Zn(II)–TPA complexes **4**, **4a** and **4b** as well as their parent ligand TPA in methanol.

The influence of the auxiliary ligands, L, was also investigated for the azido compounds, **1**, **3** and **4a**. As illustrated in Figure 6, all the azido complexes when excited at their second or higher wavelength absorption band displayed luminescence emission peaks associated with LCT transitions (Figure 6, UV spectra). These complexes show bathochromic shifts consistent with their corresponding electronic absorption spectra. Therein, Complex **1** showed the largest bathochromic shift (Δλ = 115 nm) with the emission peak maximum centered at 375 nm ($λ_{ex}$ = 260 nm). Comparison of the absolute emission intensities' wavelengths and UV–Vis. spectra of the dicyanamido Zn(II) complexes **2**, **5**, **6**, **7** and **8** containing different coordinating ligands in methanol was performed (Figure S7, SM). Therein, complex **5** shows the strongest visible fluorescence peak at 421 nm ($λ_{ex}$ = 330 nm). The UV–Vis. spectrum **5** (Figure S7-Right, SM) exhibits a broad absorption peak centered around 330 nm, which is assigned to the n–π* transition of the amino group and another absorption peak in the UV region at 230 nm ascribed to the π–π* transition of the naphthalene system. The fluorescence quantum yield (FL QY) of the complex is estimated to be around 9% and close to the quantum yield (QY) of 1,8-diaminonaphthalene ligand in methanol [73]. The absolute emission intensities of **2** and **7**, seen in the UV region, roughly determined to be ~6.8- and ~9-fold, respectively, are lower than the reference complex **5** (Figure S7–FL, SM). The complexes [Zn(8-amq)$_2$(dca)]ClO$_4$ (**6**) and *catena*-[Zn(N,N-Me$_2$en)(μ$_{1,5}$-dca)]dca (**8**) did not show any detectable luminescence emission spectra (Figure S7) because of the fluorescence quenching of the 8-amq in polar protic solvents due to hydrogen bonding with the MeOH [74] in the former complex and the lack of π–π* transition in the *N,N*-Me$_2$en of Complex **8**.

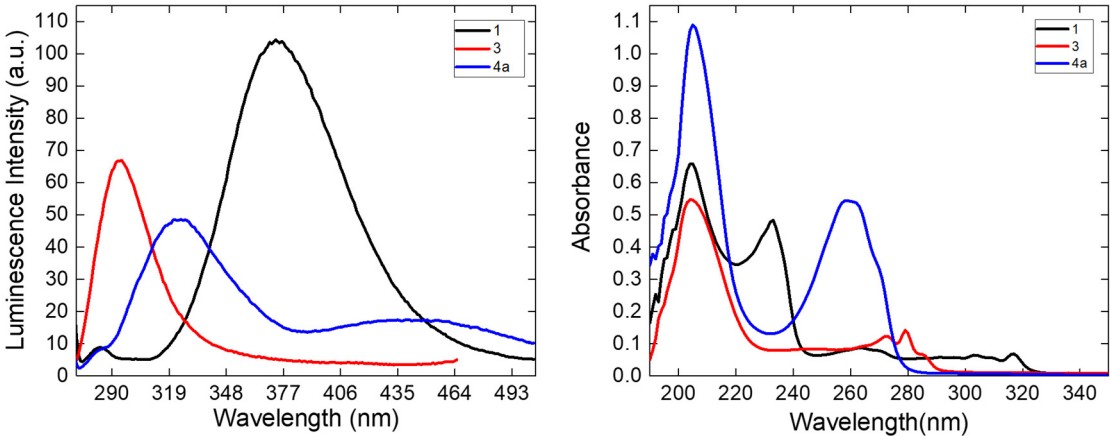

**Figure 6.** Left: luminescence emission (λex = 260 nm), and Right: UV–Vis. absorption spectra of Zn(II)–azido complexes **1**, **3** and **4a** in methanol.

The fluorescence quantum yield (FL QY) of complex **5** is estimated to be around 9% and close to the QY of 1,8-diaminonaphthalene ligand in methanol [73]. The absolute emission intensities of **2** and **7**, seen in the UV region, roughly determined to be ~6.8- and ~9-fold, respectively, are lower than the reference complex **5** (Figure S7—FL, SM). The complexes [Zn(8-amq)₂(dca)]ClO₄ (**6**) and *catena*-[Zn(*N*,*N*-Me₂en)(μ₁,₅-dca)]dca (**8**) did not show any detectable luminescence emission spectra (Figure S7) because of the fluorescence quenching of the 8-amq in polar protic solvents due to hydrogen bonding with the MeOH [74] in the former complex and the lack of π–π* transition in the *N*,*N*-Me₂en of complex **8**. The FL QY of the pseudohalido-Zn(II) complexes under investigation in MeOH together with λmax absorption and emission is tabulated in Table 3. The FL QY of all complexes was measured in MeOH by an optically dilute relative method using 1,8 diaminonaphthalene (1,8-damnaph) as a fluorescent standard of known QY in methanol [73].

**Table 3.** The λmax absorption, λmax emission, and FL QY of the pseudohalido-Zn(II) complexes under investigation in MeOH.

| Complex | λmax Absorption or λex (nm) | λem Emission (nm) | FL QY |
|---|---|---|---|
| [Zn(Meepmqa)(N₃)₂] (**1**) | 260 | 373 | 2.62% |
| [Zn(Meepmqa)(dca)]ClO₄·½H₂O (**2**) | 260 | 373 | 1.15% |
| [Zn(NTB)(N₃)]ClO₄·½H₂O (**3**) | 240 | 294 | 0.98% |
| [Zn(TPA)(NCS)]ClO₄ (**4**) | 260 | 450 | 0.67% |
| [Zn(TPA)(N₃)]ClO₄ (**4a**) | 260 | 324, 450 | 1.50% |
| [Zn(TPA)(NCS)]NO₃·½H₂O (**4b**) | 260 | 450 | 0.50% |
| [Zn(1,8-damnph)₂(dca)₂] (**5**) | 330 | 421 | 9.00% |
| [Zn(8-amq)₂(dca)]ClO₄ (**6**) | 260 | 323 | 0.25% |
| [Zn(isq)₂(μ₁,₅-dca)₂] (**7**) | 260 | 335 | 0.56% |

The luminescence emission observed in the complexes **3–6a** may be attributed to the CHE effect, which tends to reduce energy loss via radiationless thermal vibrations, and the intraligand π*–π emission band may shift due to perturbations in the electronic states of the ligands upon coordination with Zn²⁺ ions [52,53]. Attempts made to correlate the fluorescence enhancements to the rigidity of chelation or to strong Zn–Nav(amine) bond lengths were unsuccessful. For example, in the case of Zn–TPA series, all Zn–Nav (TPA):

2.111 (**4**), 2.111 (**4a**) [68] and 2.094 Å (**4b**) [72] were very close to providing a satisfactory explanation for the fluorescence enhancement differences between the three TPA complexes, and in some cases the lack of crystal structures as compounds **1** and **2**.

## 4. Conclusions

In this study, the interaction of Zn(II) perchlorate or nitrate with pseudohalides and in the presence of *N*-donor auxiliary ligands resulted in the formation of a series of mononuclear five-coordinate complexes, including $[Zn(NTB)(N_3)]ClO_4 \cdot \frac{1}{2}H_2O$ (**3**), $[Zn(TPA)(NCS)]ClO_4$ (**4**) and two distorted octahedral $[Zn(1,8\text{-damnph})_2(dca)_2]$ (**5**) and $[Zn(8\text{-amq})_2(dca)_2]$ (**6a**) as well as two 1D polymeric chains *catena*-$[Zn(isq)_2(\mu_{1,5}\text{-dca})_2]$ (**7**) and *catena*-$[Zn(N,N\text{-Me}_2en)_2(\mu_{1,5}\text{-dca})]dca$ (**8**). The compounds $[Zn(TPA)(X)]ClO_4$ (X = $NCS^-$, $N_3^-$, dca) display luminescence emission maxima in methanol at ~ 450 nm with a large red shift up to $\Delta\lambda = 190$ nm. The effect of different auxiliary ligands, L, on the emission spectra examined in the azido and dicyanamido series revealed that the strongest visible fluorescence peak at 421 nm ($\lambda_{ex}$ = 330 nm) was observed in $[Zn(1,8\text{-damnph})_2(dca)_2]$ (**5**) and, as expected, there was no detectable luminescence emission in **8** due to the lack of $\pi-\pi^*$ nor **6** due to fluorescence quenching in MeOH [74]. The comparison of our series together and the structurally characterized other related Zn(II)-pseudohalido complexes with the corresponding Cd(II) analogues clearly showed that the prediction of the Zn(II) structure based on a structurally characterized Cd(II) analogue or vice versa is not an easy query.

**Supplementary Materials:** The following are available online at www.mdpi.com/2304-6740/9/7/53/s1, Figure S1: Packing plots of **3**, Figure S2: Packing plots of **4**, Figure S3: Packing plot of **5**, Figure S4: Packing plot of **6a**, Figure S5: Packing plot of **7**, Figure S6: Packing plot of **8**, Figure S7: Fluorescence and UV-Vis. spectra of the dicyanamido Zn(II) complexes **2**, **5**, **6**, **7** and **8** in methanol, Table S1: Selected bond distances (Å) and bond angles (°) of **3**, **4**, **5**, **6a**, **7** and **8**, Table S2: Possible hydrogen bonds, Table S3: ring·ring interactions in **7**. CCDC 2089669–2089674 contain the supplementary crystallographic data for **3, 4, 5, 6a**.

**Author Contributions:** F.A.M., R.C.F. and A.T. performed the X-ray structural analysis. A.R.D., S.F.A., N.M.H.S., F.R.L. and S.S.M. contributed to the synthesis and spectral characterization of the compounds. S.P.S., F.R.L. and S.S.M. in studying the luminescence properties of the complexes. F.A.M., N.M.H.S., F.R.L. and S.S.M. contributed to the writing of the manuscript. All authors have read and agreed to the published version of the manuscript.

**Funding:** Financial support of this work came from the Department of Chemistry at UL Lafayette by F.R.L. and S.S.M.

**Acknowledgments:** S.S.M. and F.R.L. thank the Department of Chemistry (UL Lafayette) for funding this work.

**Conflicts of Interest:** The authors declare no conflict of interest.

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
