# Peer review of "Stereochemical Geometries and Photoluminescence in Pseudo-Halido-Zinc(II) Complexes. Structural Comparison between the Corresponding Cadmium(II) Analogs"

_inorganics, doi:10.3390/inorganics9070053_

Round 1
Reviewer 1 Report
This manuscript is suitable for publication after some required revisions.
-Integrated in Scheme 1, a numbering of all investigated complexes would be useful.
-I suggest to switch paragraph 3.5 with 3.4.
-A Table reporting λ(max) absorption, λ(max) emission, and FLQY for all investigated complexes would be useful.
-Figure 6: please, use the same color for absorption and emission spectra of 1, 3, and 4a.
-In Zn(II) d10 metal complexes there are not LMCT transitions!
Author Response
Review 1
(x) I would not like to sign my review report
( ) I would like to sign my review report
English language and style
( ) Extensive editing of English language and style required
(x) Moderate English changes required
( ) English language and style are fine/minor spell check required
( ) I don't feel qualified to judge about the English language and style
Yes |
Can be improved |
Must be improved |
Not applicable |
|
Does the introduction provide sufficient background and include all relevant references? |
(x) |
( ) |
( ) |
( ) |
Is the research design appropriate? |
(x) |
( ) |
( ) |
( ) |
Are the methods adequately described? |
(x) |
( ) |
( ) |
( ) |
Are the results clearly presented? |
( ) |
(x) |
( ) |
( ) |
Are the conclusions supported by the results? |
( ) |
(x) |
( ) |
( ) |
Comments and Suggestions for Authors
This manuscript is suitable for publication after some required revisions.
-Integrated in Scheme 1, a numbering of all investigated complexes would be useful.
It is not clear to what the reviewer wants us to do here because all complexes were numbered, and all ligands used in this work were abbreviated as shown in Scheme 1.
-I suggest to switch paragraph 3.5 with 3.4.
We absolutely agree as this is a good point and switch between the two paragraphs was made. Thank you,
-A Table reporting λ(max) absorption, λ(max) emission, and FLQY for all investigated complexes would be useful.
Thank you for paying our attention to this point, we have included Table 3 (p-19) reporting λ(max) absorption, λ(max) emission, and FLQY for all investigated luminescent complexes (please see SM section . The FL QY of all complexes in methanol was measured by an optically dilute relative method using 1,8 diaminonaphthalene as fluorescent standard of known QY in methanol [From Ref 70].
-Figure 6: please, use the same color for absorption and emission spectra of 1, 3, and 4a.
Thank you for careful revision and colors were adjusted and corrected.
-In Zn(II) d10 metal complexes there are not LMCT transitions!
The reviewer is correct about this point and as a result we modified the description on page 11 of the manuscript to be as follows:
As seen in the Figure 5, upon photoexcitation at 260 nm, complexes 4, 4a and 4b showed pronounced emission maxima at ~ 450 nm with large red shift up to ∆λ = 190 nm was observed, which can be ascribed to ligand sensitized charge transfer transition (LCT) from the lowest excited energy states within the whole complex.

Reviewer 2 Report
This paper treat series of Zn(II) complexes and Cd(II) complexes which are newly synthesized by the authors.. Please consult with other reviewers for the novelty and importance of the synthesis. About the spectroscopic part, I have following comments.
At first, absorption spectra need to be shown. After that, we can discuss about the emission spectra.
Figure 5 left is normalized. However, there are no standards for that of 4a and 4b.
Concentrations of the solutions are required. If the concentrations are much highrer, re-absorption may disturb the emission intensity at shorter wavelength region.
Figure 6. Colors of the spectra between absorption and emission are not consistent.
Author Response
Review 2
(x) I would not like to sign my review report
( ) I would like to sign my review report
English language and style
( ) Extensive editing of English language and style required
( ) Moderate English changes required
( ) English language and style are fine/minor spell check required
(x) I don't feel qualified to judge about the English language and style
Yes |
Can be improved |
Must be improved |
Not applicable |
|
Does the introduction provide sufficient background and include all relevant references? |
(x) |
( ) |
( ) |
( ) |
Is the research design appropriate? |
(x) |
( ) |
( ) |
( ) |
Are the methods adequately described? |
( ) |
(x) |
( ) |
( ) |
Are the results clearly presented? |
( ) |
(x) |
( ) |
( ) |
Are the conclusions supported by the results? |
(x) |
( ) |
( ) |
( ) |
Comments and Suggestions for Authors
This paper treat series of Zn(II) complexes and Cd(II) complexes which are newly synthesized by the authors. Please consult with other reviewers for the novelty and importance of the synthesis. About the spectroscopic part, I have following comments.
At first, absorption spectra need to be shown. After that, we can discuss about the emission spectra.
The absorption and emission spectra were given in Figs: 5, 6 & S7
Figure 5 left is normalized. However, there are no standards for that of 4a and 4b.
We apologize for this confusion and would like to clarify the FL results presented in Figure 5 (left).
As described in the manuscript, it is noted from normalized emission intensities that complex 4a with N3- exhibited the strongest luminescence at ~ 450 nm among the TPA series (Figure 5, left). Therefore, we normalized the 4a with observed peak maximum at 450 nm and compared it to those of 4 and 4b. The luminescence of 4a is found to be nearly 1.6 times greater than the visible luminescence of complex 4b with dca and about 1.18-fold higher than luminescence intensity of 4.
Concentrations of the solutions are required. If the concentrations are much highrer, re-absorption may disturb the emission intensity at shorter wavelength region.
The FL intensities of all complexes and QY were determined in dilute solutions and matching absorbance values with optical density (O.D.) of less than 0.2 using methanol as solvent. The concentration of all solutions was kept constant at O.D ~ 0.2 or less at desired excitation wavelengths usually measured at λexc.= 260 nm.
We have added the following sentence in the Materials and Physical measurements (p-4)
The FL QY of all complexes in methanol was measured by an optically dilute (Optical Density < 0.2) relative method using 1,8 diaminonaphthalene as fluorescent standard of known QY in methanol.
Figure 6. Colors of the spectra between absorption and emission are not consistent.
We Fixed the color as indicated. We apologize for this mistake.
Round 2
Reviewer 1 Report
The revised version is satisfactory.
Reviewer 2 Report
The manuscript was well revised.